# Sex-Specific Catabolic Metabolism Alterations in the Critically Ill following High Dose Vitamin D

**DOI:** 10.3390/metabo12030207

**Published:** 2022-02-25

**Authors:** Sowmya Chary, Karin Amrein, Sherif H. Mahmoud, Jessica A. Lasky-Su, Kenneth B. Christopher

**Affiliations:** 1Biogen, Inc., Cambridge, MA 02142, USA; sowmyachary24@gmail.com; 2Division of Endocrinology and Diabetology, Medical University of Graz, 8036 Graz, Austria; karin.amrein@medunigraz.at; 3Faculty of Pharmacy and Pharmaceutical Sciences, University of Alberta, Edmonton, AB T6G 2R3, Canada; smahmoud@ualberta.ca; 4Channing Division of Network Medicine, Brigham and Women’s Hospital, Harvard Medical School, Boston, MA 02115, USA; jessica.a.su@gmail.com; 5Division of Renal Medicine, Brigham and Women’s Hospital, Harvard Medical School, Boston, MA 02115, USA

**Keywords:** sex, metabolomics, women, vitamin D, acylcarnitine, bioenergesis

## Abstract

Pharmacological interventions are essential for the treatment and management of critical illness. Although women comprise a large proportion of the critically ill, sex-specific pharmacological properties are poorly described in critical care. The sex-specific effects of vitamin D_3_ treatment in the critically ill are not known. Therefore, we performed a metabolomics cohort study with 1215 plasma samples from 428 patients from the VITdAL-ICU trial to study sex-specific differences in the metabolic response to critical illness following high-dose oral vitamin D_3_ intervention. In women, despite the dose of vitamin D_3_ being higher, pharmacokinetics demonstrated a lower extent of vitamin D_3_ absorption compared to men. Metabolic response to high-dose oral vitamin D_3_ is sex-specific. Sex-stratified individual metabolite associations with elevations in 25(OH)D following intervention showed female-specific positive associations in long-chain acylcarnitines and male-specific positive associations in free fatty acids. In subjects who responded to vitamin D_3_ intervention, significant negative associations were observed in short-chain acylcarnitines and branched chain amino acid metabolites in women as compared to men. Acylcarnitines and branched chain amino acids are reflective of fatty acid B oxidation, and bioenergesis may represent notable metabolic signatures of the sex-specific response to vitamin D. Demonstrating sex-specific pharmacometabolomics differences following intervention is an important movement towards the understanding of personalized medicine.

## 1. Introduction

Women and men have dissimilar pharmacokinetics and pharmacodynamics in response to drug interventions [1]. In metabolism, sex-specific genetic differences are present [2,3]. Sizable studies on circulating metabolites in ambulatory patients show notable sex-specific differences [2,3,4]. Sex-specific pharmacological properties are poorly described and most drugs do not have specific dosage recommendations for women and men [5,6,7].

Critically ill patients have a profound disruption of systemic homeostasis, as reflected by circulating metabolites. Blood-based metabolomic studies early in critical illness can reflect dysregulated cellular byproducts, inform illness severity and be used to predict outcomes [8,9]. Metabolomic profiles are shown to differ in critically ill patients with and without low 25(OH)D levels [10]. Sex-specific differences are noted in 25(OH)D levels and immunomodulatory effects as well as cardiometabolic traits in ambulatory adults [11,12]. Though the pharmacokinetics is well described in outpatients, the sex-specific effects of high-dose vitamin D_3_ in the critically ill is not known [13,14,15,16].

Metabolomics before and after intervention can furnish mechanistic insight into drug response and severity of critical illness [17,18]. Sex-specific pharmacokinetics is likely present with oral medication use [19]. But, sex-specific pharmacometabolomics is poorly described, especially in critical care. Therefore, we studied sex-specific differences in the metabolic response to critical illness following high-dose oral vitamin D_3_. We hypothesize that significant sex-specific pharmacokinetic and metabolomic responses to vitamin D_3_ intervention exist.

We performed a post hoc metabolomics cohort study of 1215 plasma samples from 428 patients collected during the VITdAL-ICU trial [20]. The VITdAL-ICU trial was a single-center trial of 492 critically ill adults with 25-hydroxyvitamin D [25(OH)D] levels ≤ 20 ng/mL randomized to high-dose oral vitamin D_3_ or placebo [20]. The VITdAL-ICU trial was negative for the primary outcome of length of stay. In a secondary outcome, the VITdAL-ICU trial showed that in patients with low baseline vitamin D levels mortality was lower in patients who received high-dose oral vitamin D_3_ compared to those who received placebo [20]. We assessed the sex-specific changes in individual metabolites following high-dose vitamin D intervention in critical illness. Secondly, we determined if sex-specific differences in pharmacokinetics or mediation exist of the relationship between the response to intervention and individual metabolite change.

## 2. Results

### 2.1. Demographics

Baseline characteristics were similar between groups of subjects stratified by absolute increase in 25(OH)D ≥ 7.5 ng/mL between day 0 and 3 and then sex for SAPS II, C-reactive protein, baseline 25(OH)D, body mass index (BMI), diabetes and ICU type. Differences existed with respect to age, Charlson Comorbidity Index, intervention status, change in 25(OH)D and day 0 levels of glucose (Table 1). The overall 28-day mortality of the 428 subject analytic cohort was 22.2%. Twenty-eight-day mortality differed based on absolute increase in 25(OH)D levels between day 0 and day 3 (χ2(1) = 13.3; *p* < 0.001) but not gender (χ2(1) = 0.014; *p* = 0.91).

### 2.2. Pharmacokinetics

We next evaluated the sex-specific pharmacokinetics of high-dose oral vitamin D_3_. The amount of vitamin D_3_ administered was the same in men and women (oral 540,000 IU vitamin D_3_); the dose of vitamin D_3_ (IU/kg), in general, is higher in women due to their lower body weight. Despite different IU/kg doses, the pharmacokinetics of 25(OH)D in patients randomized to vitamin D_3_ showed similar mean serum 25(OH)D concentrations over time (Appendix A). Compared with women, the pharmacokinetic parameters of 25(OH)D using non-compartmental analysis showed significantly higher normalized AUC0-7 in men (*p* < 0.05), a measure of the extent of drug absorption.

### 2.3. Sex-Stratified Analyses

In day 0 plasma samples of female subjects (N = 151), no significant differences were found in any individual metabolites by *t*-test (all *q*-values > 0.10) (Appendix A) or in metabolomic profiles by OPLS-DA (CV-ANOVA, *p* = 0.99) in subjects with an increase of 25(OH)D< or ≥7.5 ng/mL from day 0 to day 3 (Appendix A). In male subjects at day 0 (N = 277), no significant differences in individual metabolites were found by *t*-test (all *q*-values > 0.10) (Appendix A) or in metabolomic profiles as determined by OPLS-DA (CV-ANOVA, *p*-value 0.99) in those with an increase in 25(OH)D ≥ 7.5 ng/mL from day 0 to 3 relative to subjects who did not (Appendix A).

In sex-stratified mixed effects modeling of day 0, 3 and 7 plasma samples in 151 women (435 total samples) and 277 men (795 total samples), sex-specific differences were present, with increases in 25(OH)D between day 0 and day 3. Significant increases in long-chain acylcarnitines with elevated 25(OH)D were seen only in the female stratum (summary of data presented in Table 2, Figure 1; full data presented in Appendix A). In the male but not the female stratum, significant negative associations were found with ceramides, dicarboxylate fatty acids and long chain fatty acids in the setting of increasing 25(OH)D (summarized data in Table 3, Figure 2; full data in Appendix A). The common response in both the female and male strata to elevations in 25(OH)D is significantly increased branched chain amino acid (BCAA) metabolism and plasmalogens as well as decreases in pentose phosphate pathway metabolites (Table 4; full data in Appendix A). In the sex-stratified mixed effects regression, additional adjustment for the Charlson Comorbidity Index did not materially alter the direction and effect size of the associations between sex and metabolite pathways (Appendix A).

### 2.4. Responder Cohort Analysis

Restricting the mixed effects modeling of day 0, 3 and 7 plasma samples to the 153-patient responder cohort (those who received high dose vitamin D_3_ intervention and had an increase in 25(OH)D ≥ 7.5 ng/mL) shows sex-specific metabolite patterns over time. Compared to men, women had lower levels of BCAA metabolites and short-chain acylcarnitines and higher levels of sphingomyelin species (Appendix A). In the mixed effects regression of the responder cohort, additional adjustment for the Charlson Comorbidity Index did not change the direction, strength or significance of the associations between sex and metabolite pathways (Appendix A).

### 2.5. Mediation Analysis

Finally, we focused on the potential mediation by sex of the relationship between metabolite abundance and absolute increase in 25(OH)D level between day 0 and day 3. Mediation analyses in day 3 data revealed no sex-dependent influence on associations between the absolute increase in 25(OH)D and all 983 metabolites (all mediation *p*-values were >0.01). As both adiposity and critical illness increase free fatty acids [21,22], we evaluated BMI as a sex-specific mediator of response to vitamin D_3_. When individually restricted to female (N = 151) or to male subjects (N = 277), mediation analyses of the day 3 data revealed no influence of BMI on associations between the absolute increase in 25(OH)D and all 983 metabolites (all mediation *p*-values were >0.01).

## 3. Discussion

Our large post hoc metabolomics study considered temporal changes in the metabolome and 25(OH)D concentration as well as vitamin D_3_ dose to determine sex-specific differences in response to vitamin D_3_ intervention. We evaluated the response to vitamin D_3_ intervention in three ways: (1) a sex-specific pharmacokinetics study; (2) sex stratification to study metabolite differences relative to the change in 25(OH)D in women separately from metabolite differences in men; and (3) a responder cohort sub-analysis to directly compare metabolite differences between women and men who received vitamin D_3_ intervention and responded with an absolute increase in 25(OH)D ≥ 7.5 ng/mL between day 0 and 3. Our approach to evaluate sex-specific responses resulted in novel findings. First, our pharmacokinetics data identified sex-specific differences in vitamin D_3_ absorption. Second, we showed a sex-specific metabolomic response to vitamin D_3_ associated with differential metabolite patterns over time early in the course of critical illness. These analyses demonstrate evidence of sex-specific metabolism following high-dose vitamin D_3_.

In the sex-stratified analysis, significant associations with long-chain acylcarnitines and increased 25(OH)D over day 0 to 3 were noted within the female stratum but not in the male stratum (Table 2, Figure 1). With increased 25(OH)D, a significantly lower abundance of ceramides, dicarboxylate fatty acids and long chain fatty acids was limited to the male stratum (Table 3, Figure 2). Furthermore, the male but not the female stratum showed significant decreases in polyunsaturated fatty acids with increased 25(OH)D (Table 3, Figure 2). The common response to increased 25(OH)D in both female and male strata was a significant increase in plasmalogens and BCAA metabolites as well as a decrease in pentose phosphate pathway metabolites (Appendix A). In the responder cohort data directly comparing metabolite differences in women and men who received and responded to high-dose vitamin D_3_ (N = 153), we note that circulating BCAA metabolites and short-chain acylcarnitines are both significantly lower in women compared to men (Appendix A).

To understand the relevance of the sex-specific metabolite patterns we observed, existing studies provide context. At homeostasis, metabolism, as well as drug pharmacokinetics and pharmacodynamics, is sex-specific [19,23,24,25,26]. In healthy subjects, circulating fatty acids are incorporated into triglycerides in women but oxidized in men [27]. Compared to men, healthy women have decreased free fatty acid-induced insulin resistance, lower circulating acylcarnitines and higher levels of sphingomyelins [3]. But, studies in healthy adults have little relevance to the critically ill. The hypercatabolism of critical illness substantially alters how oxidative fuel is produced from amino acids, fatty acids and carbohydrates [28]. The physiological stress of critical illness alters such energy metabolism changes in a sex-specific fashion [29]. Drug pharmacodynamics and pharmacokinetics variability is shown in critical illness but it is not yet known to be sex-specific [30].

In the sex-stratified analysis, we found that critically ill women differed from men in the patterns of change of free fatty acids with response to vitamin D_3_. Specifically, with an increase in 25(OH)D, circulating free fatty acids were lower and more significant within the male stratum (Figure 2). White adipose tissue is prominent in regulating metabolic homeostasis [31]. In healthy adults, sex-specific differences exist in the adipocyte cytokines adiponectin and leptin, with higher levels in women relative to men [32,33]. Experimental models show that vitamin D has a complex active role in adipose tissue metabolism by modifying inflammation and adipocyte secretory function [34]. Small human studies note that ceramide, the highly bioactive lipid mediator, may be regulated by vitamin D [35,36]. Further study is needed to untangle the mechanism for the free fatty acid elevations noted in women responsive to vitamin D.

In the sex-stratified analysis, similar metabolomic patterns of change in the female and male strata are noted with higher levels of BCAA metabolites and plasmalogens and decreased levels of pentose phosphate pathway metabolites in response to increased 25(OH)D levels. These metabolite pathways are involved in bioenergesis, redox regulation and endothelial protection, respectively. During inflammation, amino acids are released into the circulation with protein breakdown. With inadequate mitochondrial fatty acid β-oxidation during critical illness, BCAAs are metabolized by the liver to high-energy compounds—a process that may be augmented by vitamin D signaling [37,38]. Plasmalogens are important antioxidants that act to protect endothelial cells from injury due to oxidative stress [39]. Pentose phosphate pathway metabolites increase with insufficient fatty acid β-oxidation, producing NADPH for redox regulation and ribose 5-phosphate for biosynthesis [40]. Our observation of lower levels of pentose phosphate pathway metabolites with increased 25(OH)D levels may be related to the induction of glutathione formation by vitamin D, which provides cell and mitochondrial protection via lower ROS levels and lower NADPH demand [10,41].

Fatty acid β-oxidation is the key pathway for fatty acid breakdown and energy production and is profoundly dysregulated in critical illness [42]. Fatty acid β-oxidation occurs in both mitochondria and peroxisomes. While long-, medium- and short-chain fatty acids are oxidized by mitochondria, very long-chain fatty acids, bile acids, branched-chain fatty acids and dicarboxylic fatty acids are oxidized by peroxisomes [43,44,45]. Vitamin D enhances fatty acid β-oxidation through upregulation of transcription factors, including PPAR 1α and the mitochondrial enzymes pyruvate dehydrogenase kinase 4, as well as carnitine-palmitoyl transferase 1a and 1b [46,47,48]. Further, data suggest that vitamin D is essential for mitochondrial oxidative phosphorylation capacity [49,50].

In the sex-stratified analysis, we found that critically ill women and men differ in the patterns of change in acylcarnitines. Specifically, higher and more significant increases in long-chain acylcarnitines were present in the female stratum in response to increased 25(OH)D levels. Mitochondria in women utilize fatty acids over amino acids for energy production, create less reactive oxygen species and have an increased oxygen capacity compared to men [51,52,53,54,55]. Production of long-chain acylcarnitines occurs when elevated fatty acid supply exceeds mitochondrial oxidative capacity [56]. In exercise and insulin resistance, such mitochondrial overload is observed, resulting in elevated circulating long-chain acylcarnitines [43,57]. The elevated long-chain acylcarnitines we observed in women are most likely due to increased fatty acid release that exceeds the oxidative capacity of mitochondria [56].

Important to the understanding of our observed long-chain acylcarnitine sex-specific response to vitamin D_3_ is the contribution of mitochondrial fatty acid uptake. The fatty acid transporter FAT/CD36 which facilitates fatty acid entry into mitochondria is shown to be more abundant in women [58,59,60]. FAT/CD36 abundance may be important for higher uptake of long-chain fatty acids by mitochondria in women [61]. Our observation of increased long-chain acylcarnitines in women relative to men following vitamin D_3_ may reflect an overwhelming of fatty acid β-oxidation by increased fatty acid delivery to mitochondria in women.

In the sex-stratified data, we find that BCAA catabolic metabolites increase in both men and women with increases in 25(OH)D levels. In our responder cohort analysis of the 153 patients who received and responded to high-dose vitamin D_3_, we observed that circulating BCAA catabolic metabolites and short-chain acylcarnitines were significantly lower in women relative to men. The relative decrease in even short-chain acylcarnitines (C4) in women is evidence of more complete fatty acid β-oxidation compared to men [43]. The C3 and C5 short-chain acylcarnitines are products of catabolism of BCAAs, threonine and methionine [62]. Our observed sex-specific differences in BCAA catabolic metabolites and short-chain acylcarnitines may reflect a greater enhancement of mitochondrial oxidative function in women with response to vitamin D_3_. Taken together, our results suggest that in the context of higher free fatty acids delivery, female mitochondria are more responsive to enhancement of mitochondrial oxidative function via vitamin D_3_, which results in a lower relative production of short-chain acylcarnitines.

Our novel study approach has several strengths. The sample size was large and the repeated measurement of subjects over time substantially increases our statistical power [37,63]. Linear mixed models are useful for determination of metabolite abundance at multiple time points [64,65]. Furthermore, we adjusted for multiple comparisons using the false discovery rate [66]. Our work has potential limitations inherent to its metabolomics cohort study design. Specifically, as an observational study it is subject to bias and may have limited causal inference. The source of samples were White patients with serum 25(OH)D ≤ 20 ng/mL, so the results are potentially not generalizable to all critically ill patients.. Incomplete adjustment for the heterogeneity of illness may exist despite adjustment for demographics and clinical data. Though expression of 25-hydroxylase is not sex-specific, our use of 25(OH)D in determining absorption does not take into account potential differences in the activation of vitamin D_3_ in the liver [67,68]. It is important to note that our study is post hoc and should be considered hypothesis generating.

## 4. Materials and Methods

Detailed methods are presented in Appendix A. In the VITdAL-ICU trial, 475 critically ill adults with serum 25-hydroxyvitamin D [25(OH)D] ≤ 20 ng/mL were randomized to oral- or nasogastric tube-administered vitamin D_3_ 540,000 IU followed by 90,000 IU monthly or placebo (oleum arachidis) [20]. Trial inclusion criteria included patients admitted to a medical or surgical ICU who were 18 years or older with an expected ICU stay of ≥48 h and a 25(OH)D level of 20 ng/mL or lower. Excluded from the trial were: patients with severely impaired gastrointestinal function; other trial participation; pregnancy; lactating women; tuberculosis; sarcoidosis; hypercalcemia; nephrolithiasis in the past year; and patients not deemed suitable for study participation (i.e., psychiatric disease, prisoner status). Blood samples were collected on days 0 (at randomization), 3 and 7. Plasma was fractionated, aliquoted and stored at −80 °C. Frozen plasma was available in 453 VITdAL-ICU trial subjects, of whom 25 were excluded due to absence of serum 25(OH)D determination at day 3.

For determination of the pharmacokinetics of oral vitamin D_3_, we utilized serum 25(OH)D levels, a marker of systemic vitamin D status [69]. Serum 25(OH)D levels were determined via chemiluminescence assay (IDS-iSYS, Immunodiagnostic Systems) with assay coefficients of variation for control material of 9.4% at 64 ng/mL, 10% at 31 ng/mL and 13.4% at 13 ng/mL [20]. For pharmacokinetics evaluation, the area under the plasma concentration–time curve from vitamin D_3_ dosing to day 7 (AUC0-7d) was calculated using the linear trapezoidal method. Patients with missing 25(OH)D levels on day 3 or 7 and those who received placebo were excluded from the pharmacokinetics evaluation. AUC normalized to vitamin D_3_ dose and body weight (AUCnorm) was calculated by dividing AUC0-7d by dose in IU per kg body weight. Median AUCnorm and serum 25(OH)D levels on days 0, 3 and 7 values were compared between males and females.

Metabolomics data was produced from 1215 plasma samples obtained from 428 patients: 428 samples were analyzed from day 0, 413 at day 3 and 374 at day 7. Four ultra-high-performance liquid chromatography/tandem accurate mass spectrometry (UHPLC/MS/MS) methods were utilized by Metabolon, Inc. (Morrisville, NC, USA) to identify 983 metabolites [18,29]. We cube root-transformed and Pareto-scaled the metabolomics data to generate an approximate normal distribution.

Based on our previous metabolomics analysis of the VITdAL-ICU trial [18], we considered a response to high-dose oral vitamin D_3_ as an absolute increase in 25(OH)D ≥ 7.5 ng/mL from day 0 to day 3. For a sex-stratified analysis of day 0 data, we utilized the Student’s *t*-test to determine the significance of individual metabolites between vitamin D_3_ response groups (25(OH)D< or ≥7.5 ng/mL from day 0 to day 3) using MetaboAnalyst in women and in men [70]. We corrected for multiple testing via the Benjamini–Hochberg procedure to adjust the false discovery rate (FDR) to 0.10, producing a *q*-value [66]. We analyzed sex-stratified day 0 data via orthogonal partial least squares-discriminant analysis (OPLS-DA), a statistical method for supervised classification discrimination (SIMCA 15.0 Umetrics, Umea, Sweden). We utilized the OPLS-DA approach to find the metabolomic differences at baseline (day 0) between patients who did and did not respond to high-dose vitamin D_3_. We analyzed male and female subjects separately to determine if such differences in metabolites at baseline were sex-specific. We used goodness of fit (R2) and predictive performance (Q2) to describe the OPLS-DA model quality. To validate the OPLS-DA model, we performed permutation testing. To determine overall OPLS-DA model significance, we used sevenfold cross-validation analysis of variance (CV-ANOVA).

For day 0, 3 and 7 repeated measures data, correlations between individual metabolites and absolute increase in 25(OH)D levels from day 0 to day 3 were determined separately in women and in men utilizing sex-stratified linear mixed effects models correcting for age, baseline 25(OH)D, SAPS II, sample day, admission diagnosis, and a subject-specific random intercept. A false discovery rate adjusted *p*-value (*q*-value) threshold of 0.10 was utilized to identify all significant differences [66]. Additionally, we performed a sub-analysis in a responder cohort of patients who received high-dose vitamin D_3_ intervention and had an increase in 25(OH)D ≥ 7.5 ng/mL from day 0 to day 3. For day 0, 3 and 7 repeated measures data, we determined correlations between individual metabolites over time relative to sex (as the exposure) using linear mixed effects models adjusted for age, baseline 25(OH)D, SAPS II, sample day, admission diagnosis and a subject-specific random intercept. All mixed effects models were analyzed using STATA 16.1 (College Station, TX, USA). For data visualization purposes, rain plots were produced in R-3.6.2 [71].

Lastly, we evaluated a potential sex-specific mediating effect on the association between the absolute rise in 25(OH)D levels from day 0 to day 3 and individual metabolite abundance, adjusting for baseline 25(OH)D, age, SAPS II and diagnosis at admission. Each of the 983 metabolites at day 3 were analyzed using mediation, an R package for causal mediation analysis [72], to determine bootstrap *p*-values (N = 2000) for the sex-specific mediating effect. Mediation was significantly present if ≥10% of the association was mediated through sex and the *p*-value was <0.01 [73].

## 5. Conclusions

In a large metabolomics study, we demonstrated substantial differences between women and men in the response to high-dose vitamin D_3_ early in critical illness. Specifically, robust sex-specific differences in pharmacokinetics and metabolomics were found in the response over time to high-dose vitamin D_3_. Demonstrating sex-specific differences in the metabolic response to pharmacologic intervention is a crucial first step in the understanding of personalized medicine.

## Figures and Tables

**Figure 1 metabolites-12-00207-f001:**
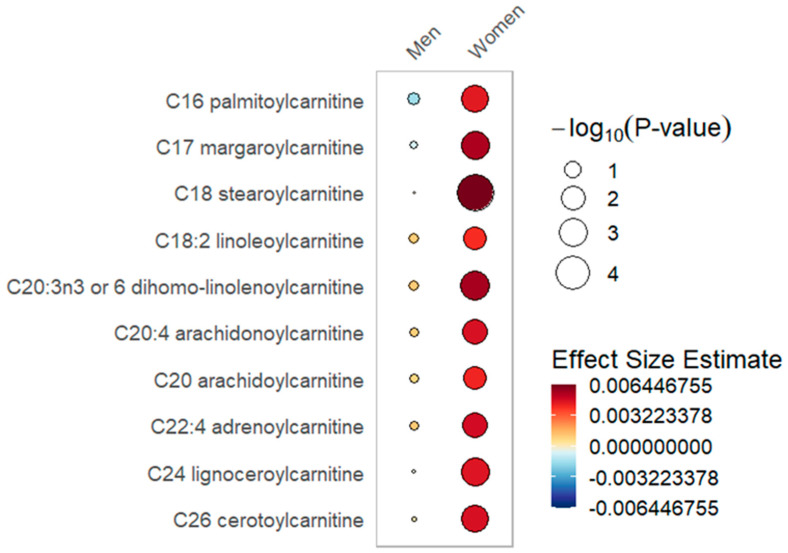
Long-chain acylcarnitines rain plot. Correlations between absolute increase in 25(OH)D levels from day 0 to day 3 and individual long-chain acylcarnitine metabolite abundance at day 0, 3 or 7 in women or men determined utilizing mixed-effects linear regression models correcting for age, SAPS II, admission diagnosis and 25(OH)D at day 0. The red color fill scare indicates the magnitude of beta coefficient estimates and the size of the circle corresponds to the significance level (−log_10_(*p*-value)). Statistical significance is the multiple test-corrected threshold of −log_10_(*p*) > 1.92 which is equivalent to *p* < 0.0122.

**Figure 2 metabolites-12-00207-f002:**
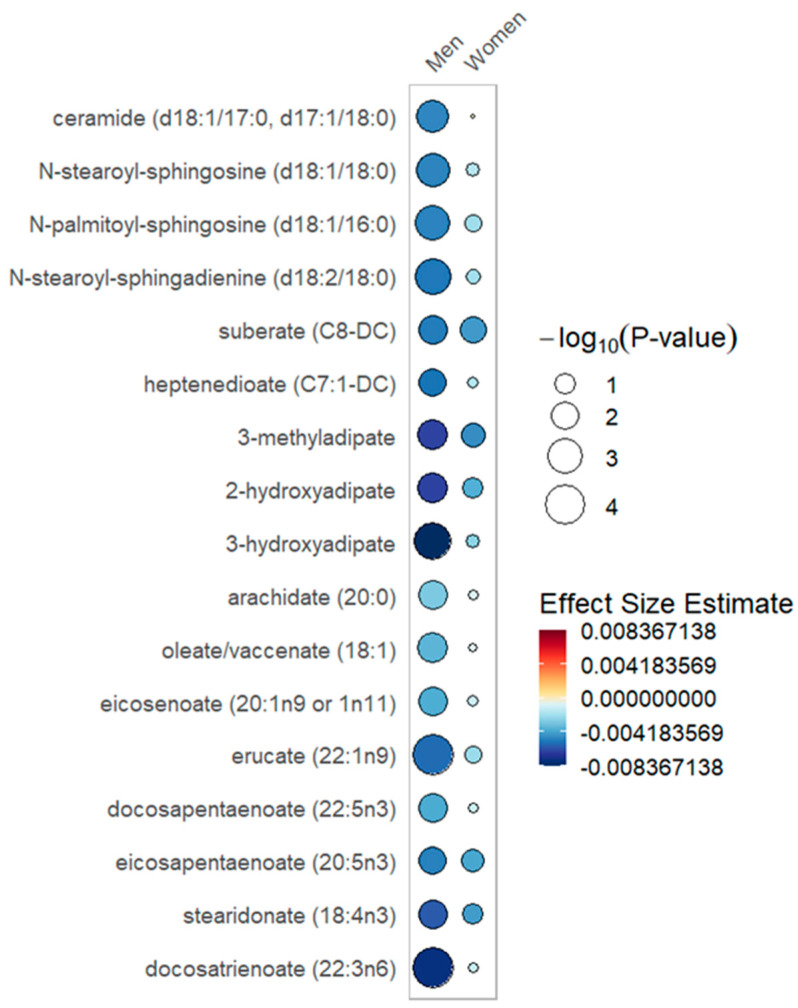
Free fatty acid metabolites rain plot. Correlations between absolute increase in 25(OH)D levels from day 0 to day 3 and individual free fatty acid metabolite abundance at day 0, 3 or 7 in women or men determined utilizing mixed-effects linear regression models correcting for age, SAPS II, admission diagnosis and 25(OH)D at day 0. The blue color fill scare indicates the magnitude of beta coefficient estimates and the size of the circle corresponds to the significance level (−log_10_(*p*-value)). Statistical significance is the multiple test-corrected threshold of −log_10_(*p*) > 1.92, which is equivalent to *p* < 0.0122.

**Table 1 metabolites-12-00207-t001:** Characteristics of analytic cohort, according to sex and 25(OH)D response.

Characteristic	Absolute Increase in 25(OH)D Level between Day 0 and Day 3				Total	*p*-Value
	25(OH)D ≥ 7.5 ng/mL		25(OH)D < 7.5 ng/mL			
	Female	Male	Female	Male		
No.	59	97	92	180	428	
Age (Years) Mean (SD)	66.5 (15.6)	63.3 (16.4)	69.3 (11.5)	61.2 (14.7)	64.2 (14.9)	<0.001 *
SAPS II Mean (SD)	34.0 (16.2)	33.0 (17.0)	34.9 (13.8)	32.5 (15.1)	33.4 (15.4)	0.67 *
CRP μg/mL Day 0 Mean (SD)	125.9 (101.8)	125.2 (93.6)	116.0 (93.2)	128.9 (81.9)	124.9 (89.8)	0.74 *
Day 0 25(OH)D ng/mL Mean (SD)	12.5 (4.5)	15.6 (15.7)	13.6 (6.4)	13.7 (4.9)	14 (8.8)	0.15 *
Vitamin D_3_ Intervention No. (%)	57 (97)	96 (99)	21 (23)	38 (21)	212 (50)	<0.001
Change in 25(OH)D ng/mL Median (IQR)	22.3 [14.7, 37.1]	22.2 [14.7, 38.2]	0.5 [−1.6, 2.5]	0.6 [−1.2, 2.9]	3.3 [0, 16.7]	<0.001 ^‡^
BMI Mean (SD)	26.7 (4.8)	27.2 (4.6)	27.5 (5.8)	26.9 (5.4)	27.1 (5.2)	0.75 *
Charlson Comorbidity Index Mean (SD)	2.2 (1.6)	3.1 (2.4)	3.2 (2.1)	3.0 (2.2)	3.0 (2.2)	<0.001 *
Diabetes History No. (%)	15 (25)	19 (20)	24 (26)	43 (24)	101 (24)	0.73
Glucose Day 0 Mean (SD)	167.7 (64.5)	147.5 (46.1)	153.5 (50.7)	144.2 (48.5)	150.3 (51.4)	0.025 *
ICU						0.13
Anesthesia ICU No. (%)	10 (17)	15 (16)	14 (15)	42 (24)	81 (19)	
Cardiac Surgery ICU No. (%)	13 (22)	24 (25)	27 (30)	59 (33)	123 (29)	
Medicine ICU No. (%)	12 (21)	19 (20)	19 (21)	39 (22)	89 (21)	
Neurological ICU No. (%)	21 (36)	30 (32)	26 (29)	29 (16)	106 (25)	
Surgical No. (%)	2 (3)	7 (7)	5 (5)	8 (5)	22 (5)	
28-day Mortality No. (%)	11 (19)	11 (11)	23 (25)	50 (28)	95 (22)	0.013

Note: Data presented as n (%) unless otherwise indicated. *p*-values determined by a chi-squared test unless designated by (*), in which case the *p*-value was determined by ANOVA, or by (^‡^) in which case the *p*-value was determined by the Kruskal–Wallis test. Vitamin D_3_ Intervention No. (%) is the number and proportion of patients who were randomized to receive Vitamin D_3_.

**Table 2 metabolites-12-00207-t002:** Long-chain acylcarnitines significantly associated with response to high-dose vitamin D in women but not men.

Metabolite	Female Stratum			Male Stratum			Sub-Pathway
	β Coefficient	*p*-Value	*q*-Value	β Coefficient	*p*-Value	*q*-Value	
Palmitoylcarnitine (C16)	0.004593	2.65 × 10^−3^	**5.83 × 10^−2^**	−0.00156	2.76 × 10^−1^	4.96 × 10^−1^	Long-chain Acylcarnitine
Margaroylcarnitine (C17) *	0.005591	1.19 × 10^−3^	**5.27 × 10^−2^**	−0.0007	6.47 × 10^−1^	7.81 × 10^−1^	Long-chain Acylcarnitine
Stearoylcarnitine (C18)	0.006447	1.08 × 10^−5^	**7.89 × 10^−3^**	0.0000581	9.67 × 10^−1^	9.81 × 10^−1^	Long-chain Acylcarnitine
Linoleoylcarnitine (C18:2) *	0.004122	1.11 × 10^−2^	**9.81 × 10^−2^**	0.001059	4.35 × 10^−1^	6.29 × 10^−1^	Long-chain Acylcarnitine
Dihomo-linolenoylcarnitine (C20:3n3 or 6) *	0.005715	7.64 × 10^−4^	**4.15 × 10^−2^**	0.001106	4.73 × 10^−1^	6.55 × 10^−1^	Long-chain Acylcarnitine
Arachidonoylcarnitine (C20:4)	0.004813	6.21 × 10^−3^	**7.59 × 10^−2^**	0.001038	5.10 × 10^−1^	6.78 × 10^−1^	Long-chain Acylcarnitine
Arachidoylcarnitine (C20) *	0.004313	1.21 × 10^−2^	**9.89 × 10^−2^**	0.000811	5.71 × 10^−1^	7.29 × 10^−1^	Long-chain Acylcarnitine
Adrenoylcarnitine (C22:4) *	0.00495	5.79 × 10^−3^	**7.59 × 10^−2^**	0.001161	4.98 × 10^−1^	6.68 × 10^−1^	Long-chain Acylcarnitine
Lignoceroylcarnitine (C24) *	0.004725	1.17 × 10^−3^	**5.27 × 10^−2^**	−0.00017	9.02 × 10^−1^	9.43 × 10^−1^	Long-chain Acylcarnitine
Cerotoylcarnitine (C26) *	0.004831	1.86 × 10^−3^	**5.50 × 10^−2^**	0.000254	8.68 × 10^−1^	9.21 × 10^−1^	Long-chain Acylcarnitine

Note: Mixed effects modeling results for selected metabolites significant in women. For day 0, 3 and 7 repeated measures data, correlations between individual metabolites and absolute increase in 25(OH)D levels from day 0 to 3 were determined separately in women (N = 151) and in men (N = 277) utilizing linear mixed effects models correcting for age, baseline 25(OH)D, absolute increase in 25(OH)D at day 3, SAPS II, sample day, admission diagnosis and an individual subject-specific random intercept. Bolded *q*-value results indicate statistical significance following false discovery rate (FDR) adjustment to 0.10. * metabolites are identified via predictive or externally acquired structure evidence when a reference standard does not exist.

**Table 3 metabolites-12-00207-t003:** Free fatty acid metabolites significantly associated with responses to high-dose vitamin D in men but not women.

Metabolite	Female Stratum			Male Stratum			Sub-Pathway
	β Coefficient	*p*-Value	*q*-Value	β Coefficient	*p*-Value	*q*-Value	
Ceramide (d18:1/17:0, d17:1/18:0) *	−0.00018	9.15 × 10^−1^	9.60 × 10^−1^	−0.00487	1.84 × 10^−3^	**4.49 × 10^−2^**	Ceramide
N-stearoyl-sphingosine (d18:1/18:0) *	−0.00159	3.62 × 10^−1^	5.96 × 10^−1^	−0.00494	1.59 × 10^−3^	**4.08 × 10^−2^**	Ceramide
N-palmitoyl-sphingosine (d18:1/16:0)	−0.00208	1.87 × 10^−1^	4.10 × 10^−1^	−0.00496	9.80 × 10^−4^	**2.95 × 10^−2^**	Ceramide
N-stearoyl-sphingadienine (d18:2/18:0) *	−0.00199	2.73 × 10^−1^	5.08 × 10^−1^	−0.00525	6.44 × 10^−4^	**2.28 × 10^−2^**	Ceramide
Suberate (C8-DC)	−0.00435	1.79 × 10^−2^	1.23 × 10^−1^	−0.00513	6.47 × 10^−3^	**7.18 × 10^−2^**	Fatty Acid, Dicarboxylate
Heptenedioate (C7:1-DC) *	−0.00161	5.08 × 10^−1^	7.24 × 10^−1^	−0.00535	1.27 × 10^−2^	**9.53 × 10^−2^**	Fatty Acid, Dicarboxylate
3-methyladipate	−0.00469	3.39 × 10^−2^	1.62 × 10^−1^	−0.00648	5.36 × 10^−3^	**6.71 × 10^−2^**	Fatty Acid, Dicarboxylate
2-hydroxyadipate	−0.00377	8.30 × 10^−2^	2.65 × 10^−1^	−0.00652	5.18 × 10^−3^	**6.65 × 10^−2^**	Fatty Acid, Dicarboxylate
3-hydroxyadipate *	−0.00247	3.31 × 10^−1^	5.68 × 10^−1^	−0.00837	3.84 × 10^−4^	**1.72 × 10^−2^**	Fatty Acid, Dicarboxylate
Arachidate (20:0)	−0.00073	5.61 × 10^−1^	7.56 × 10^−1^	−0.00287	7.31 × 10^−3^	**7.28 × 10^−2^**	Long Chain Fatty Acid
Oleate/vaccenate (18:1)	−0.00063	6.90 × 10^−1^	8.42 × 10^−1^	−0.00357	6.20 × 10^−3^	**7.18 × 10^−2^**	Long Chain Fatty Acid
Eicosenoate (20:1n9 or 1n11)	−0.00109	5.16 × 10^−1^	7.28 × 10^−1^	−0.00383	6.73 × 10^−3^	**7.28 × 10^−2^**	Long Chain Fatty Acid
Erucate (22:1n9)	−0.00221	1.72 × 10^−1^	3.90 × 10^−1^	−0.00553	5.62 × 10^−5^	**7.48 × 10^−3^**	Long Chain Fatty Acid
Docosapentaenoate (22:5n3)	−0.00108	5.60 × 10^−1^	7.55 × 10^−1^	−0.00391	9.29 × 10^−3^	**8.36 × 10^−2^**	Polyunsaturated Fatty Acid
Eicosapentaenoate (20:5n3)	−0.00402	5.46 × 10^−2^	2.10 × 10^−1^	−0.00503	1.07 × 10^−2^	**8.94 × 10^−2^**	Polyunsaturated Fatty Acid
Stearidonate (18:4n3)	−0.00428	8.68 × 10^−2^	2.70 × 10^−1^	−0.00595	7.82 × 10^−3^	**7.63 × 10^−2^**	Polyunsaturated Fatty Acid
Docosatrienoate (22:3n6) *	−0.00118	5.86 × 10^−1^	7.67 × 10^−1^	−0.00759	6.13 × 10^−5^	**7.48 × 10^−3^**	Polyunsaturated Fatty Acid

Note: Mixed effects modeling results for selected metabolites significant in men. For day 0, 3 and 7 repeated measures data, correlations between individual metabolites and absolute increase in 25(OH)D levels from day 0 to 3 were determined separately in women (N = 151) and in men (N = 277) utilizing linear mixed effects models correcting for age, baseline 25(OH)D, absolute increase in 25(OH)D at day 3, SAPS II, sample day, admission diagnosis, and an individual subject-specific random intercept. Bolded *q*-value results indicate statistical significance following false discovery rate (FDR) adjustment to 0.10. * metabolites are identified via predictive or externally acquired structure evidence when a reference standard does not exist.

**Table 4 metabolites-12-00207-t004:** Metabolites significantly associated with response to a high dose of vitamin D in both women and men.

Metabolite	Female Stratum			Male Stratum			Sub-Pathway
	β Coefficient	*p*-Value	*q*-Value	β Coefficient	*p*-Value	*q*-Value	
3-methyl-2-oxovalerate	0.00454	4.82 × 10^−4^	**3.05 × 10^−2^**	0.00372	2.47 × 10^−3^	**5.24 × 10^−2^**	BCAA Metabolism
4-methyl-2-oxopentanoate	0.00357	1.02 × 10^−2^	**9.48 × 10^−2^**	0.00370	3.78 × 10^−3^	**5.98 × 10^−2^**	BCAA Metabolism
Leucine	0.00293	4.96 × 10^−3^	**7.24 × 10^−2^**	0.00372	2.41 × 10^−4^	**1.34 × 10^−2^**	BCAA Metabolism
Isoleucine	0.00293	5.01 × 10^−3^	**7.24 × 10^−2^**	0.00313	3.32 × 10^−4^	**1.62 × 10^−2^**	BCAA Metabolism
1-(1-enyl-stearoyl)-2-arachidonoyl-GPE (P-18:0/20:4) *	0.00708	1.61 × 10^−5^	**7.89 × 10^−3^**	0.00563	1.17 × 10^−4^	**1.16 × 10^−2^**	Plasmalogen
1-(1-enyl-stearoyl)-2-linoleoyl-GPE (P-18:0/18:2) *	0.00559	2.93 × 10^−4^	**2.38 × 10^−2^**	0.00514	1.66 × 10^−5^	**5.39 × 10^−3^**	Plasmalogen
1-(1-enyl-palmitoyl)-2-linoleoyl-GPE (P-16:0/18:2) *	0.00551	2.03 × 10^−3^	**5.50 × 10^−2^**	0.00693	3.95 × 10^−6^	**2.14 × 10^−3^**	Plasmalogen
1-(1-enyl-palmitoyl)-2-arachidonoyl-GPE (P-16:0/20:4) *	0.00484	5.14 × 10^−4^	**3.05 × 10^−2^**	0.00438	2.48 × 10^−4^	**1.34 × 10^−2^**	Plasmalogen
1-(1-enyl-palmitoyl)-2-arachidonoyl-GPC (P-16:0/20:4) *	0.00465	2.45 × 10^−3^	**5.76 × 10^−2^**	0.00522	1.20 × 10^−4^	**1.16 × 10^−2^**	Plasmalogen
1-(1-enyl-palmitoyl)-2-linoleoyl-GPC (P-16:0/18:2) *	0.00448	4.58 × 10^−3^	**7.01 × 10^−2^**	0.00434	2.47 × 10^−3^	**5.24 × 10^−2^**	Plasmalogen
1-(1-enyl-palmitoyl)-2-palmitoyl-GPC (P-16:0/16:0) *	0.00414	2.43 × 10^−4^	**2.22 × 10^−2^**	0.00414	2.43 × 10^−4^	**1.34 × 10^−2^**	Plasmalogen
Ribitol	−0.00464	1.01 × 10^−2^	**9.48 × 10^−2^**	−0.00395	6.71 × 10^−3^	**7.28 × 10^−2^**	Pentose Metabolism
Gluconate	−0.00940	5.22 × 10^−3^	**7.40 × 10^−2^**	−0.01015	5.30 × 10^−4^	**2.15 × 10^−2^**	Pentose Metabolism
Arabitol/xylitol	−0.00628	2.01 × 10^−3^	**5.50 × 10^−2^**	−0.00463	6.97 × 10^−3^	**7.28 × 10^−2^**	Pentose Metabolism

Note: Mixed effects modeling results of selected metabolites significant in both women and men. For day 0, 3 and 7 repeated measures data, correlations between individual metabolites and absolute increase in 25(OH)D levels from day 0 to 3 were determined separately in women (N = 151) and in men (N = 277) utilizing linear mixed effects models correcting for age, baseline 25(OH)D, absolute increase in 25(OH)D at day 3, SAPS II, sample day, admission diagnosis, and an individual subject-specific random intercept. Bolded *q*-value results indicate statistical significance following false discovery rate (FDR) adjustment to 0.10. * metabolites are identified via predictive or externally acquired structure evidence when a reference standard does not exist.

## Data Availability

The data presented in this study are available in Appendix A.

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
