# Peer review of "Sex-Specific Catabolic Metabolism Alterations in the Critically Ill following High Dose Vitamin D"

_metabolites, 2022, doi:10.3390/metabo12030207_

Round 1

Reviewer 1 Report

The manuscript Sex-specific catabolic metabolism alterations in the critically ill following High Dose Vitamin D, indicates that the sex-specific pharmacometabolomics differences following intervention are an important step towards the understanding of personalized medicine.

The introduction is very short

The manuscript is descriptive. It is needed the inclusion of the relationship of Vitamin D with fatty acid oxidation, BCAA metabolites, ceramide metabolism,…..

What are the authors referring to when they speak of critically ill patients? Do all the pathologies that patients have a common molecular mechanism? The patients have diseases with a sufficiently varied etiology to be able to be associated in a study.

Is there a matched group of healthy subjects?

“In a secondary outcome, the VITdAL-ICU trial showed that in patients with low baseline 25(OH)D levels, mortality improved with high dose oral vitamin D3”.  This phrase is confusing, since it is not known if mortality improves, more patients die, or there is a decrease in mortality.

“We next evaluated sex-specific pharmacokinetics of high dose oral vitamin D3. Though the dose of vitamin D3 (IU/kg) is higher in women, the pharmacokinetics of 25(OH)D in patients randomized to vitamin D3 showed similar mean serum 25(OH)D concentrations over time”. I cannot find in the Materials and Methods Section, what is the dose of vitD used or if it is different between sexes.

Discussion of the manuscript should be written more clearly, and variations in different metabolites: acyl carnitines and BCAA metabolites should be unified.

The manuscript Sex-specific catabolic metabolism alterations in the critically ill following High Dose Vitamin D, indicates that the sex-specific pharmacometabolomics differences following intervention are an important step towards the understanding of personalized medicine.

The introduction is very short

The manuscript is descriptive. It is needed the inclusion of the relationship of Vitamin D with fatty acid oxidation, BCAA metabolites, ceramide metabolism,…..

What are the authors referring to when they speak of critically ill patients? Do all the pathologies that patients have a common molecular mechanism? The patients have diseases with a sufficiently varied etiology to be able to be associated in a study.

Is there a matched group of healthy subjects?

“In a secondary outcome, the VITdAL-ICU trial showed that in patients with low baseline 25(OH)D levels, mortality improved with high dose oral vitamin D3”.  This phrase is confusing, since it is not known if mortality improves, more patients die, or there is a decrease in mortality.

“We next evaluated sex-specific pharmacokinetics of high dose oral vitamin D3. Though the dose of vitamin D3 (IU/kg) is higher in women, the pharmacokinetics of 25(OH)D in patients randomized to vitamin D3 showed similar mean serum 25(OH)D concentrations over time”. I cannot find in the Materials and Methods Section, what is the dose of vitD used or if it is different between sexes.

Discussion of the manuscript should be written more clearly, and variations in different metabolites: acylcarnitines and BCAA metabolites should be unified.

Author Response

Thank you for your detailed constructive criticism regarding our work. We sincerely appreciate your time and efforts. We have now addressed each point raised and provided clarification, correction or new data where requested. Our point-by-point response comments follow “Authors' Response.”

The manuscript Sex-specific catabolic metabolism alterations in the critically ill following High Dose Vitamin D, indicates that the sex-specific pharmacometabolomics differences following intervention are an important step towards the understanding of personalized medicine.

The introduction is very short

Authors’ Response: The introduction is concise but has been expanded by 97 words.

The manuscript is descriptive. It is needed the inclusion of the relationship of Vitamin D with fatty acid oxidation, BCAA metabolites, ceramide metabolism,…..

Authors’ Response: Yes, as noted in the manuscript the study is a post-hoc metabolomics cohort study. We have previously published on the relationship between the metabolomic response to high dose vitamin D in the VITdAL-ICU trial [1].

We now include the following in the discussion

“Vitamin D enhances fatty acid β-oxidation through upregulation of transcription factors including PPAR 1α and the mitochondrial enzymes pyruvate dehydrogenase kinase 4 as well as carnitine-palmitoyl transferase 1a and 1b [2-4].”

“BCAAs are metabolized by the liver to high-energy compounds a process that may be augmented by vitamin D signaling [5,6].”

“Small human studies note that ceramide, the highly bioactive lipid mediator may be regulated by vitamin D [7,8].”

What are the authors referring to when they speak of critically ill patients?

Authors’ Response: Critically ill patients are those who either suffer from life-threatening conditions or are at risk of developing them and are cared for in intensive care units specialized to provide life sustaining measures such as mechanical ventilation.

Do all the pathologies that patients have a common molecular mechanism?

Authors’ Response: The common molecular mechanism of severe critical illness is not known. It is under investigation [9] but appears to be related to energy utilization pathways [10].

The patients have diseases with a sufficiently varied etiology to be able to be associated in a study.

Authors’ Response: Evolutionary forces create enormous genetic heterogeneity in human illness which is magnified in critical illness [11]. Our large number of patients with repeated sampling combined with the mixed effects regression analysis and subanalysis allow for adjustment for the etiology of critical illness. Including covariates in the mixed effects model provides the potential to explain some of the heterogeneity. We included the following patient level covariates in our mixed effects model: age, severity of illness, admission diagnosis, baseline 25(OH)D, absolute increase in 25(OH)D at day 3, plasma day sample was measured, and an individual subject-specific random-intercept. This is a powerful approach to utilize data to inform estimation of individual patient covariate effects on the sex-specific metabolomic response to high dose vitamin D3. We have included the following in the limitations: “Incomplete adjustment for the heterogeneity of illness may exist despite adjustment for demographics and clinical data.”

Is there a matched group of healthy subjects?

Authors’ Response: The plasma analyzed is from the VITdAL-ICU trial, a clinical trial of critically ill patients who are admitted to an intensive care unit and randomized to receive high dose vitamin D3 or placebo [12]. Healthy patients were not enrolled in the VITdAL-ICU trial thus there is not a matched group of healthy subjects. Metabolic homeostasis is known to be profoundly disturbed in the critically ill compared to healthy subjects [13].

“In a secondary outcome, the VITdAL-ICU trial showed that in patients with low baseline 25(OH)D levels, mortality improved with high dose oral vitamin D3”.  This phrase is confusing, since it is not known if mortality improves, more patients die, or there is a decrease in mortality.

Authors’ Response: Thank you for your comment. We have rewritten the sentence for clarity as

“In a secondary outcome, the VITdAL-ICU trial showed that in patients with low baseline vitamin D levels, mortality was lower in patients who received high dose oral vitamin D3 compared to those who received placebo.”

“We next evaluated sex-specific pharmacokinetics of high dose oral vitamin D3. Though the dose of vitamin D3 (IU/kg) is higher in women, the pharmacokinetics of 25(OH)D in patients randomized to vitamin D3 showed similar mean serum 25(OH)D concentrations over time”. I cannot find in the Materials and Methods Section, what is the dose of vitD used or if it is different between sexes.

Authors’ Response: The dose of vitamin D3 given was the same in men and women: oral 540,000 IU cholecalciferol. The dose per kilogram differed from patient to patient as the kilogram weight in general differed between men and women.

We have clarified the text to read “The amount of vitamin D3 administered was the same in men and women (oral 540,000 IU vitamin D3), the dose of vitamin D3 (IU/kg) in general is higher in women due to lower kg weight. Despite different IU/kg doses, the pharmacokinetics of 25(OH)D in patients randomized to vitamin D3 showed similar mean serum 25(OH)D concentrations over time”

Discussion of the manuscript should be written more clearly, and variations in different metabolites: acylcarnitines and BCAA metabolites should be unified.

Authors’ Response: Metabolism in critical illness is profoundly disturbed along multiple pathways. To understand the data produced in our study, we separately discussed the known evidence of those pathways that appeared to be most differential between women and men. Regarding acylcarnitines and BCAA metabolites, we have noted that the C3 and C5 short-chain acylcarnitines are products of catabolism of BCAAs. We have also added 110 words to the discussion to increase clarity.

References Cited

  1. Amrein, K.; Lasky-Su, J.A.; Dobnig, H.; Christopher, K.B. Metabolomic basis for response to high dose vitamin D in critical illness. Clinical nutrition 2020, 10.1016/j.clnu.2020.09.028,10.1016/j.clnu.2020.09.028.
  2. Yin, Y.; Yu, Z.; Xia, M.; Luo, X.; Lu, X.; Ling, W. Vitamin D attenuates high fat diet-induced hepatic steatosis in rats by modulating lipid metabolism. European journal of clinical investigation 2012, 42, 1189-1196,10.1111/j.1365-2362.2012.02706.x.
  3. Marcotorchino, J.; Tourniaire, F.; Astier, J.; Karkeni, E.; Canault, M.; Amiot, M.J.; Bendahan, D.; Bernard, M.; Martin, J.C.; Giannesini, B., et al. Vitamin D protects against diet-induced obesity by enhancing fatty acid oxidation. J Nutr Biochem 2014, 25, 1077-1083,10.1016/j.jnutbio.2014.05.010.
  4. Sergeev, I.N.; Song, Q. High vitamin D and calcium intakes reduce diet-induced obesity in mice by increasing adipose tissue apoptosis. Molecular nutrition & food research 2014, 58, 1342-1348,10.1002/mnfr.201300503.
  5. Kobayashi, H.; Amrein, K.; Lasky-Su, J.; Christopher, K.B. Procalcitonin Metabolomics in the Critically Ill reveal relationships between inflammation intensity and energy utilization pathways. Scientific Reports 2021, 11, 23194,10.1038/s41598-021-02679-0.
  6. Dimitrov, V.; Barbier, C.; Ismailova, A.; Wang, Y.; Dmowski, K.; Salehi-Tabar, R.; Memari, B.; Groulx-Boivin, E.; White, J.H. Vitamin D-regulated Gene Expression Profiles: Species-specificity and Cell-specific Effects on Metabolism and Immunity. Endocrinology 2021, 162,10.1210/endocr/bqaa218.
  7. Chen, L.; Dong, Y.; Bhagatwala, J.; Raed, A.; Huang, Y.; Zhu, H. Vitamin D3 Supplementation Increases Long-Chain Ceramide Levels in Overweight/Obese African Americans: A Post-Hoc Analysis of a Randomized Controlled Trial. Nutrients 2020, 12,10.3390/nu12040981.
  8. Koch, A.; Grammatikos, G.; Trautmann, S.; Schreiber, Y.; Thomas, D.; Bruns, F.; Pfeilschifter, J.; Badenhoop, K.; Penna-Martinez, M. Vitamin D Supplementation Enhances C18(dihydro)ceramide Levels in Type 2 Diabetes Patients. Int J Mol Sci 2017, 18,10.3390/ijms18071532.
  9. Kobayashi, H.; Amrein, K.; Lasky-Su, J.A.; Christopher, K.B. Procalcitonin metabolomics in the critically ill reveal relationships between inflammation intensity and energy utilization pathways. Sci Rep 2021, 11, 23194,10.1038/s41598-021-02679-0.
  10. Langley, R.J.; Migaud, M.E.; Flores, L.; Thompson, J.W.; Kean, E.A.; Mostellar, M.M.; Mowry, M.; Luckett, P.; Purcell, L.D.; Lovato, J., et al. A metabolomic endotype of bioenergetic dysfunction predicts mortality in critically ill patients with acute respiratory failure. Sci Rep 2021, 11, 10515,10.1038/s41598-021-89716-0.
  11. McClellan, J.; King, M.C. Genetic heterogeneity in human disease. Cell 2010, 141, 210-217,10.1016/j.cell.2010.03.032.
  12. Amrein, K.; Schnedl, C.; Holl, A.; Riedl, R.; Christopher, K.B.; Pachler, P.; Urbanic-Purkart, T.; Waltensdorfer, A.; Münch, A.; Warnkross, H., et al. Effect of high dose Vitamin D3 on hospital length of stay in critically ill patients with Vitamin D Deficiency: A Randomized Clinical Trial. JAMA 2014, Epub September 30,10.1001.
  13. Kiehntopf, M.; Nin, N.; Bauer, M. Metabolism, metabolome, and metabolomics in intensive care: is it time to move beyond monitoring of glucose and lactate? Am J Respir Crit Care Med 2013, 187, 906-907,10.1164/rccm.201303-0414ED.

Reviewer 2 Report

Thanks for the opportunity to revise this paper proposed by Dr Chary and colleagues.

I really appreciate this so interesting and extensive work conducted by the authors. Since several findings are available regarding vitamin d metabolism gender-differences, the topic of this work is of great and current interest.

The paper is well written and methods and results clearly reported.

I have no specific revisions to apply for the authors. I only suggest to include in introduction section further evidences regarding gender-disparities on vitamin d metabolism and it’s relationships with cardiometabolic features (DOI: 10.1159/000458765) and critical inflammatory/infectious illness (DOI: 10.1210/clinem/dgab599).

Author Response

Thank you for your constructive criticism regarding our work. We sincerely appreciate your time and efforts. We have now addressed each point raised and provided clarification, correction or new data where requested. Our point-by-point response comments follow “Authors' Response.”

Thanks for the opportunity to revise this paper proposed by Dr Chary and colleagues.

I really appreciate this so interesting and extensive work conducted by the authors. Since several findings are available regarding vitamin d metabolism gender-differences, the topic of this work is of great and current interest.

The paper is well written and methods and results clearly reported.

I have no specific revisions to apply for the authors. I only suggest to include in introduction section further evidences regarding gender-disparities on vitamin d metabolism and it’s relationships with cardiometabolic features (DOI: 10.1159/000458765) and critical inflammatory/infectious illness (DOI: 10.1210/clinem/dgab599).

Authors’ Response: Thank you for your suggestion. We have now included the following in the introduction “Sex-specific differences are noted in 25(OH)D levels and immunomodulatory effects as well as cardiometabolic traits in ambulatory adults.”

Reviewer 3 Report

The work by Chary et al. is interesting, well done and presents a new approach to analyzing the effects of vitamin D, by evaluating the metabolic differences between males and females. In fact, the study analyzes a cohort of critically ill patients and searches for the correlation between the effects of supplementation with vitamin D and the variation of some metabolic parameters, depending on the gender.

I have some doubts about the figures and I do not fully agree with some conclusions reached by the authors.

Figure 1. Unclear meaning of “ Vitamin D3 Intervention No. (%)”.

Figure 3 is mentioned in the text, but the figure is not shown (Lines 240-242).

Table 3 and Figure 2: The authors state “significant positive associations exist with ceramides, dicarboxylate fatty acids, and long  chain fatty acids in the setting of increasing 25(OH)D”. It seems negative association. Also discussion needs to be revised accordingly.

Discussion:

  1. The authors conclude that “pharmacokinetics data identified sex-specific differences in vitamin D3 absorption”. If they refer to the lower values ​​of 25OH vit D, it is not necessarily a different absorption, but perhaps a different activation of the precursor in the liver. Are there any data about the gender-dependent expression of 25-hydroxylase?

  1. When a metabolite is high, the reason can be either low utilization or increased production. If long chain acylcarnitine is higher in women, this could be due to higher mobilized fat or lower utilization by beta-oxidation. In fact, the carnitine-bound acyl is transferred to mitochondrial compartment and released to fuel beta-oxidation. If this catabolic pathway is slow, the entry is decreased and acylcarnitine accumulates. The authors seem to reach the opposite conclusion, because they say that vitamin D potentiate beta-oxidation and they state “ Mitochondrial fatty acid β-oxidation produces the majority of circulating long-chain acylcarnitines [50]”. The reference is cited with a wrong meaning, the cited article compares peroxisomal and mitochondrial utilization of fatty acids, and means that long-chain acyls are conjugated to carnitine in mitochondrial metabolism. The increased long chain acylcarnitine observed in women in this study could be explained by an increased release of fatty acids exceeding mitochondrial oxidative capacity. On the other hand, short chain acylcarnitine are lower in women probably due to mitochondrial stimulation by vitamin D, which enhances the oxidative catabolism of BCAA, as the authors suggest.

  1. The authors should consider the antioxidant effect of vitamin D. In fact, the hormone induces the synthesis of glutathione, protecting the whole cell (and mitochondria as well) from ROS production. This could be the reason why pentose phosphate pathway is decreased when vitamin D is higher (Table 4), due to the lower ROS levels and lower NADPH demand.

Author Response

Thank you for your detailed constructive criticism regarding our work. We sincerely appreciate your time and efforts. We have now addressed each point raised and provided clarification, correction or new data where requested. Our point-by-point response comments follow “Authors' Response.”

The work by Chary et al. is interesting, well done and presents a new approach to analyzing the effects of vitamin D, by evaluating the metabolic differences between males and females. In fact, the study analyzes a cohort of critically ill patients and searches for the correlation between the effects of supplementation with vitamin D and the variation of some metabolic parameters, depending on the gender.

I have some doubts about the figures and I do not fully agree with some conclusions reached by the authors.

Figure 1. Unclear meaning of “ Vitamin D3 Intervention No. (%)”.

Authors’ Response: In Table 1, Vitamin D3 Intervention No. (%) is the number and proportion (%) of patients in each column who received the Vitamin D3 intervention. This is now included in the table

Figure 3 is mentioned in the text, but the figure is not shown (Lines 240-242).

Authors’ Response: My apologies, thank you for finding the error. Lines 240-242 refer to Figure 2 which is now corrected.

Table 3 and Figure 2: The authors state “significant positive associations exist with ceramides, dicarboxylate fatty acids, and long chain fatty acids in the setting of increasing 25(OH)D”. It seems negative association. Also discussion needs to be revised accordingly.

 Authors’ Response: My apologies, thank you for spotting the error. Both results and discussion sections were revised as it is a negative association.

Discussion:

  1. The authors conclude that “pharmacokinetics data identified sex-specific differences in vitamin D3 absorption”. If they refer to the lower values ​​of 25OH vit D, it is not necessarily a different absorption, but perhaps a different activation of the precursor in the liver. Are there any data about the gender-dependent expression of 25-hydroxylase?

 Authors’ Response: Thank you for your insightful comment and question. Yes, the pharmacokinetics “absorption” determination is based on serum 25(OH)D. Pharmacokinetics drug “absorption” is determined by the dose-normalized AUC [area under the plasma concentration–time curve of 25(OH)D from vitamin D3 dosing to day 7 (AUC0-7d) normalized to vitamin D3 dose and body weight (AUCnorm)]. Our determination of dose using 25(OH)D may reflect absorption as well as potential differential activation of vitamin D3 in the liver. As you know, bioactivation of vitamin D3 involves 25-hydroxylation in the liver and subsequent 1α-hydroxylation to produce 1α,25-dihydroxyvitamin D3. The CYP2R1 gene encodes the enzyme 25-hydroxylase. CYP2R1 is highly conserved across vertebrates [1] and is not sex-specific [2,3]. The CYP27B1 gene encodes 25-hydroxyvitamin D 1α hydroxylase. Sex-specific expression is not found in rodents but there is no information in humans [4]. Evidence does exist for sex-specific expression of extra-renal 25-hydroxyvitamin D 1α hydroxylase and is found to be higher in women [5]. Such extra-renal 25-hydroxyvitamin D 1α hydroxylase is thought not to contribute to circulating active vitamin D3 levels [5].

We have added the following to the limitations in the discussion “Though expression of 25-hydroxylase is not sex-specific, our use of 25(OH)D in determining absorption does not take into account potential differences in the activation of vitamin D3 in the liver [2,3].”

  1. When a metabolite is high, the reason can be either low utilization or increased production. If long chain acylcarnitine is higher in women, this could be due to higher mobilized fat or lower utilization by beta-oxidation. In fact, the carnitine-bound acyl is transferred to mitochondrial compartment and released to fuel beta-oxidation. If this catabolic pathway is slow, the entry is decreased and acylcarnitine accumulates. The authors seem to reach the opposite conclusion, because they say that vitamin D potentiate beta-oxidation and they state “Mitochondrial fatty acid β-oxidation produces the majority of circulating long-chain acylcarnitines [50]”. The reference is cited with a wrong meaning, the cited article compares peroxisomal and mitochondrial utilization of fatty acids, and means that long-chain acyls are conjugated to carnitine in mitochondrial metabolism. The increased long chain acylcarnitine observed in women in this study could be explained by an increased release of fatty acids exceeding mitochondrial oxidative capacity. On the other hand, short chain acylcarnitine are lower in women probably due to mitochondrial stimulation by vitamin D, which enhances the oxidative catabolism of BCAA, as the authors suggest.

Authors’ Response: Your point is well taken and we appreciate your argument. Though circulating acylcarnitines levels increase with the stress of critical illness, their precise role in energy metabolism and their regulation are not clear [6].  The review we cited “L-Carnitine and Acylcarnitines: Mitochondrial Biomarkers for Precision Medicine” notes the following:

“Peroxisomal β-oxidation is mostly involved with fatty acid biosynthesis, whereas mitochondrial β-oxidation is directed toward energy production [7]. Due to this functional distinction, mitochondrial β-oxidation likely produces the majority of the medium- and long-chain ACs measured in the plasma [7-10]. Medium- and long-chain ACs are produced when fatty acid supply exceeds demand and/or the capacity of mitochondrial β-oxidation and the TCA cycle enzymes [11].”

We have now omitted the above statement “Mitochondrial fatty acid β-oxidation produces the majority of circulating long-chain acylcarnitines” and use the following later in the paper:  “The elevated long-chain acylcarnitines we observe in women are most likely due to increased fatty acid release that exceeds the oxidative capacity of mitochondria [11].”

  1. The authors should consider the antioxidant effect of vitamin D. In fact, the hormone induces the synthesis of glutathione, protecting the whole cell (and mitochondria as well) from ROS production. This could be the reason why pentose phosphate pathway is decreased when vitamin D is higher (Table 4), due to the lower ROS levels and lower NADPH demand.

Authors’ Response: Thank you for your thoughtful and constructive comment. We have now included the following in the discussion regarding glutathione:

“Our observation of lower pentose phosphate pathway metabolites with increase in 25(OH)D levels may be related to the induction of glutathione formation by vitamin D which provides cell and mitochondrial protection via lower ROS levels and lower NADPH demand [12] [13].”

References Cited

  1. Nelson, D.R. Comparison of P450s from human and fugu: 420 million years of vertebrate P450 evolution. Archives of biochemistry and biophysics 2003, 409, 18-24,10.1016/s0003-9861(02)00553-2.
  2. Cheng, J.B.; Motola, D.L.; Mangelsdorf, D.J.; Russell, D.W. De-orphanization of cytochrome P450 2R1: a microsomal vitamin D 25-hydroxilase. J Biol Chem 2003, 278, 38084-38093,10.1074/jbc.M307028200.
  3. Yamasaki, T.; Izumi, S.; Ide, H.; Ohyama, Y. Identification of a novel rat microsomal vitamin D3 25-hydroxylase. J Biol Chem 2004, 279, 22848-22856,10.1074/jbc.M311346200.
  4. Krementsov, D.N.; Asarian, L.; Fang, Q.; McGill, M.M.; Teuscher, C. Sex-Specific Gene-by-Vitamin D Interactions Regulate Susceptibility to Central Nervous System Autoimmunity. Frontiers in immunology 2018, 9, 1622,10.3389/fimmu.2018.01622.
  5. Somjen, D.; Katzburg, S.; Hendel, D.; Sharon, O.; Posner, G.H. Age- and sex-dependent hormonal modulation of the expression of vitamin D receptor and 25-hydroxyvitamin D 1alpha hydroxylase in human bone cells. Expert Rev Endocrinol Metab 2014, 9, 283-292,10.1586/17446651.2014.899897.
  6. Simcox, J.; Geoghegan, G.; Maschek, J.A.; Bensard, C.L.; Pasquali, M.; Miao, R.; Lee, S.; Jiang, L.; Huck, I.; Kershaw, E.E., et al. Global Analysis of Plasma Lipids Identifies Liver-Derived Acylcarnitines as a Fuel Source for Brown Fat Thermogenesis. Cell Metab 2017, 26, 509-522 e506,10.1016/j.cmet.2017.08.006.
  7. Demarquoy, J.; Le Borgne, F. Crosstalk between mitochondria and peroxisomes. World J Biol Chem 2015, 6, 301-309,10.4331/wjbc.v6.i4.301.
  8. Schooneman, M.G.; Vaz, F.M.; Houten, S.M.; Soeters, M.R. Acylcarnitines: reflecting or inflicting insulin resistance? Diabetes 2013, 62, 1-8,10.2337/db12-0466.
  9. Palmitoylcarnitine, CID=461. Availabe online: https://pubchem.ncbi.nlm.nih.gov/compound/Palmitoylcarnitine (accessed on 2022).
  10. Bene, J.; Hadzsiev, K.; Melegh, B. Role of carnitine and its derivatives in the development and management of type 2 diabetes. Nutr Diabetes 2018, 8, 8,10.1038/s41387-018-0017-1.
  11. Bruls, Y.M.; de Ligt, M.; Lindeboom, L.; Phielix, E.; Havekes, B.; Schaart, G.; Kornips, E.; Wildberger, J.E.; Hesselink, M.K.; Muoio, D., et al. Carnitine supplementation improves metabolic flexibility and skeletal muscle acetylcarnitine formation in volunteers with impaired glucose tolerance: A randomised controlled trial. EBioMedicine 2019, 49, 318-330,10.1016/j.ebiom.2019.10.017.
  12. Lasky-Su, J.; Dahlin, A.; Litonjua, A.A.; Rogers, A.J.; McGeachie, M.J.; Baron, R.M.; Gazourian, L.; Barragan-Bradford, D.; Fredenburgh, L.E.; Choi, A.M.K., et al. Metabolome alterations in severe critical illness and vitamin D status. Crit Care 2017, 21, 193,10.1186/s13054-017-1794-y.
  13. Jain, S.K.; Micinski, D. Vitamin D upregulates glutamate cysteine ligase and glutathione reductase, and GSH formation, and decreases ROS and MCP-1 and IL-8 secretion in high-glucose exposed U937 monocytes. Biochem Biophys Res Commun 2013, 437, 7-11,10.1016/j.bbrc.2013.06.004.

Round 2

Reviewer 1 Report

The manuscript can be now accepted in the present form. The authors have answered most of my concerns

Reviewer 3 Report

The study is interesting and well written. The text has been revised as suggested and the raised issues  have been addressed. The work is now ready for publication.